# The Synergistic Effect of Interventional Locoregional Treatments and Immunotherapy for the Treatment of Hepatocellular Carcinoma

**DOI:** 10.3390/ijms24108598

**Published:** 2023-05-11

**Authors:** Nicolò Brandi, Matteo Renzulli

**Affiliations:** Department of Radiology, IRCCS Azienda Ospedaliero-Universitaria di Bologna, Via Albertoni 15, 40138 Bologna, Italy; matteo.renzulli@aosp.bo.it

**Keywords:** hepatocellular carcinoma, immunotherapy, liver, liver cancer, immune checkpoint inhibitors, tyrosine kinase inhibitors, locoregional treatment, chemoembolization, TACE, tumor ablation, radioembolization

## Abstract

Immunotherapy has remarkably revolutionized the management of advanced HCC and prompted clinical trials, with therapeutic agents being used to selectively target immune cells rather than cancer cells. Currently, there is great interest in the possibility of combining locoregional treatments with immunotherapy for HCC, as this combination is emerging as an effective and synergistic tool for enhancing immunity. On the one hand, immunotherapy could amplify and prolong the antitumoral immune response of locoregional treatments, improving patients’ outcomes and reducing recurrence rates. On the other hand, locoregional therapies have been shown to positively alter the tumor immune microenvironment and could therefore enhance the efficacy of immunotherapy. Despite the encouraging results, many unanswered questions still remain, including which immunotherapy and locoregional treatment can guarantee the best survival and clinical outcomes; the most effective timing and sequence to obtain the most effective therapeutic response; and which biological and/or genetic biomarkers can be used to identify patients likely to benefit from this combined approach. Based on the current reported evidence and ongoing trials, the present review summarizes the current application of immunotherapy in combination with locoregional therapies for the treatment of HCC, and provides a critical evaluation of the current status and future directions.

## 1. The New Era in Hepatocellular Carcinoma Treatment: The Breakthrough of Immunotherapy

Hepatocellular carcinoma (HCC) is the sixth most commonly occurring cancer worldwide, and due to its constantly increasing incidence, it has become the third leading cause of cancer-related death among general populations. Moreover, it represents the most common cause of death in patients with cirrhosis [1,2]. Multiple classification schemes are available to stratify HCC patients in an effort to determine which therapies they can undergo to increase their overall survival. The Barcelona Clinic Liver Cancer Staging (BCLC) system is one of the most widely used, and takes into account hepatic function, the extent of tumor involvement, and performance status [3].

The definitive therapies for HCC remain surgical resection and liver transplantation that can be performed only in patients at very early (0) and early (A) stages. However, given the similar survival benefit paired with the less invasiveness and lower costs compared to surgical resection, percutaneous ablative therapies such as radiofrequency ablation (RFA) and microwave ablation (MWA) are now considered the first treatment approach in both very early and early stages, especially in patients with small HCC (≤3 cm) [4,5]. Despite inducing an effective local antitumor effect, the responses to ablation techniques are relatively weak and might not completely control the tumor, as testified to by the high local recurrence rates. In particular, the size, number and location of tumors can be responsible for incomplete treatment response [6]; in addition, by promoting angiogenesis of residual cancer cells through both transcriptional and epigenetic regulations, insufficient ablation could lead to the recurrence of HCC with a more aggressive phenotype [7]. Therefore, novel techniques to improve ablation efficacy are currently being investigated.

Despite the improvement in screening and surveillance programs, most patients with HCC (about 65–70%) are still diagnosed in the intermediate (B) or advanced (C) tumoral stages, and are thus ineligible for radical therapies [8,9,10]; therefore, patients with intermediate and/or advanced HCCs are considered for transarterial therapies or systemic therapies [11,12] which, albeit effective, are deemed non-curative or “palliative” and still yield a lower 5-year survival rate [13,14]. According to BCLC tumor staging and management [3], transarterial chemoembolization (TACE) is recommended as first-line therapy for unresectable intermediate-stage HCC (stage B). Therefore, it is not surprising that this treatment was the most widely used first line treatment for the treatment of HCC across the world. More interestingly, instead, TACE emerged as the most frequently used first-line treatment for early and advanced stages, thus making it the most frequent treatment for HCC overall [8]. In fact, TACE is potentially suitable and safe for selected patients in the advanced stage with tumor vein thrombosis [15,16,17], or in combination with systemic therapies, without safety concerns [18,19]. Additionally, TACE can be safely and effectively performed in patients at very early and early stages that are partial responders to surgery or ablation, or that are unfit for these curative therapies due to contraindications [20], or prior to liver transplantation to downstage the tumor burden [21]. Despite evidence of beneficial short-term outcomes with locoregional treatments, recurrence and distant metastasis continue to have a significant effect on the overall survival of patients with HCC, especially in intermediate and advanced stages. This may be partly explained by the hypoxic environment created by the TACE procedure, which can induce neoangiogenesis by stimulating vascular endothelial growth factor (VEGF) and other angiogenic pathways, promoting revascularisation and growth of residual viable tumors or even new lesions [22]. Moreover, when it comes to transarterial therapies, one important consideration is that the blockade of hepatic arteries, especially if repeated several times, can compromise liver function and lead to collateral vessel formation, thus limiting the ability to repeat embolization by conventional hepatic vasculature [23,24]. In an effort to address this problem, many studies have been conducted combining TACE with systemic anti-angiogenic agents, most commonly sorafenib, with the aim to counteract this paradoxical effect and thus extend the clinical benefit derived from TACE. However, although most of these studies report the safety of the combination [18], a large number of clinical trials have failed to demonstrate any significant improvement in clinically relevant outcomes for patients with intermediate-stage HCC [25,26]. Even the more recent TACTICS trial, despite being the first study to demonstrate a longer progression-free survival (PFS) in patients receiving sorafenib plus TACE than in those receiving TACE plus placebo [27], it did not significantly extend overall survival (OS) in its final post-hoc analysis [28]. Therefore, better strategies to improve the outcomes for HCC patients treated with TACE are being developed.

Along with TACE, the role of other radiological locoregional therapies has expanded in recent years. For example, transarterial radioembolization (TARE) with yttrium-90 has been suggested as a safe and effective alternative treatment option for HCC patients with a liver-dominant disease who cannot tolerate systemic therapies [29,30,31], even with a significant cost advantage [32]. Moreover, the recent availability of new microspheres with a different radioisotope (such as 166-holmium) and the new technological developments will probably contribute to further reinforce the role of this option in HCC treatment and expand its clinical indication even in early and intermediate stages [33,34]. Nonetheless, due to the current lack of evidence demonstrating its superiority and non-inferiority to sorafenib, TARE is now recommended only in single HCCs ≤ 8 cm [3,35], and its role behind this indication remains uncertain.

Therefore, despite current limitations, the role of interventional radiology in the treatment of HCC is continuing to grow at each stage of the disease, especially at centers of excellence with multidisciplinary tumor boards, whether it is performed with curative, downstaging, bridging, debulking or palliative intent (Figure 1) [36,37]. Moreover, its expansion is expected to further progress as technical and clinical innovation continue to outpace large randomized controlled trials, with 50–60% of HCC patients that are expected to receive these treatments in their lifespan, globally [38].

In recent years, immunotherapy has led to a major shift in the treatment of HCC and prompted clinical trials, with therapeutic agents being used to selectively target immune cells rather than cancer cells [39]. In particular, the combination of atezolizumab and bevacizumab is now regarded as the standard first-line treatment for patients with advanced HCC due to the significant and clinically meaningful improvements in terms of OS, PFS, objective response rate (ORR) and complete response rate (CRR) compared with sorafenib monotherapy [3,40,41]. More recently, the combination of tremelimumab and durvalumab has been reported to be superior to sorafenib in patients with advanced or unresectable HCC, adding another first-line treatment option [42]. The impressive benefit provided by immunotherapy in patients with advanced HCC has led to the question if there is a rationale to support the combination of these new drugs with locoregional therapies in an adjuvant or neoadjuvant setting even in the early and/or intermediate stages [43]. In fact, it has now been demonstrated that locoregional treatments can positively alter the immune microenvironment of HCC and, theoretically, have a synergistic effect, further enhancing antitumor immune responses and thus improving patient survival [44]. In addition, novel immunotherapies, including new target antibodies, bispecific antibodies, combination regimens, engineered cytokines, adoptive T-cell therapy, tumor vaccines, and oncolytic viruses might be available to treat all stages of HCC in the near future. Based on the current reported evidence and ongoing trials, immunotherapy, especially in combination with other therapies, has the potential to act as a significant approach to the treatment of HCC.

## 2. The Immunogenic Proprieties of Hepatocellular Carcinoma

HCC arises almost exclusively in the setting of chronic liver diseases, and chronic inflammation is now regarded as one of the main triggers of hepatocarcinogenesis [45,46,47]. Since the background of chronic inflammation promotes immune suppression, there is a tightly interwoven, exceedingly complex relationship between HCC and the anti-tumor immune response in the liver. Due to the presence of an immune-suppressed microenvironment, HCC is indeed considered an immunogenic tumor [48].

First of all, chronic inflammation plays a key role in the initiation, evolution, and progression of neoplasms by creating a microenvironment that supports the malignant transformation of hepatocytes through hepatocellular DNA damage and genetic and epigenetic aberrations [49]. When liver damage occurs, thanks to the liver’s unique considerable ability to repair itself, differentiated hepatocytes can re-enter the cell cycle and serve as their own main source of replacement [50]. However, the chronic activation of non-parenchymal cells induces altered survival and proliferation signals, resulting in cellular stress, epigenetic modifications, mitochondrial alterations, DNA damage, senescence, and chromosomal aberrations. This leads to continual cell death, compensatory regeneration and liver fibrosis, which collectively induce tumorigenesis [51]. Moreover, the increased production of pro-inflammatory cytokines occurring in the setting of chronic inflammation promotes the expression of pro-oncogenic transcription factors (such as STAT3 and NF-κB), further contributing to HCC development [52].

Secondly, chronic inflammation can boost tumor immunogenicity, creating an immunosuppressive surrounding and allowing cancer cells to escape the host immune surveillance and progress [53]. One of the main functions of the liver is to continuously remove a large and diverse spectrum of pathogen components [i.e., pathogen-associated molecular patterns (PAMP)] and endogen molecules derived from damaged or necrotized cells [i.e., damage-associated molecular patterns (DAMPs)] from the circulation, thus ensuring organ protection by maintaining immunotolerance [54]. In chronic liver diseases, however, this tightly controlled immunological network is deregulated, thus leading to the failure of efficient detection and the elimination of transformed cells and causing the breakdown of proper tolerance [53]. Once HCC has developed, an intra-tumor infiltration by lymphocytes occurs, in an attempt by the host to mediate an anti-tumor reaction [55]. Under normal circumstances, tumor antigens would be internalized by the host antigen-presenting cells (APCs) and then, after being processed, be bonded to Major Histocompatibility Complex II (MHC-II) molecules. Subsequently, if properly stimulated, dendritic cells would present these tumor antigens to T cells located in the lymphatic organs, thus promoting their activation and the stimulation of effector cells, including CD8+ T cells and Natural Killer (NK) cells. Once activated, tumor-specific effector cells would migrate from lymph nodes to the tumor location, where they would exert their cytotoxic effect on neoplastic cells. Unfortunately, these cellular responses can be dysfunctional and unable to efficiently eliminate cancer cells, thus leading to HCC progression [56].

Tumoral cells can indeed promote an elevated production of immunosuppressor cytokines (such as IL-10 and TGFβ1) that downregulate the anti-tumor response at different levels. The number of immunosuppressive cells such as myeloid-derived suppressor cells (MDSCs) and regulatory T cells (Tregs) increases in the HCC microenvironment, which directly inhibits the tumor killing effect of NK cells and CD8+ T cells through overexpression of multiple factors [57]. In addition, MCH II is often functionally depleted in HCC, thus being unable to induce the activation of CD8+ T cells and leading to tumor immune escape [58]. Furthermore, tumoral cells inhibit the activation of APCs and promote the M2 polarization of macrophages, thus further impairing the effector functions of CD8+ T cells and NK cells [59,60]. Lastly, there is an abnormal expression and function of immune checkpoint molecules that, rather than preventing the excessive immune response from injuring normal hepatocytes as it happens in normal conditions, inhibit the host immune function and thus promote the growth of tumor cells. In particular, the most studied of them are programmed cell death protein 1 (PD-1) and its ligand (PD-L1), which leads to the T-cell exhaustion status, and cytotoxic T-lymphocyte protein 4 (CTLA-4), which inhibits the activation of T cells [61,62].

The current combined strategy of immunotherapy and locoregional treatments essentially aims to enhance the effects of immune checkpoint inhibitors (ICIs) that selectively target these immune checkpoints (PD-1/PD-L1 and CTLA-4); therefore, rather than stimulating new or different immune responses, ICIs can restore and unleash a preexisting immune reactivity to cancer which is being held in check by tumoral microenvironmental factors (Figure 2) [63,64].

### 2.1. PD-1 and PD-L1 Inibithors

PD-1 plays a key role in the regulation and maintenance of the balance between T cell activation and immune tolerance, especially in peripheral tissues. It is widely expressed on human cells but is mainly detected in activated T cells, NK cells and APCs [65].

PD-1 has two ligands, PD-L1 and PD-L2. PD-L1 is expressed on a variety of cells, both hematopoietic and non-hematopoietic, whereas PD-L2 is selectively induced on fewer cells post-activation, especially APCs, and has a higher affinity to PD1 than PD-L1; however, recent studies have shown that PD-L2 can be found also on other immune cells and even on tumor cells under microenvironment stimulation [66]. When PD-1 binds to its ligands on T cells, it leads to dephosphorylation of T cell receptors and blockage of CD28 signaling with subsequent reduction of T cell proliferation, adhesion, cytolytic function and cytokine production [67,68]. Reportedly, it also promotes the differentiation, maintenance and function of Tregs, further enhancing tumor immune escape [69].

In the HCC microenvironment, PD-L1 is highly expressed by intra-tumoral inflammatory cells, especially Kupffer cells and other APCs, which thereby prevent the activation of anti-tumor T cells [70,71]. Additionally, tumor cells can turn this immune checkpoint signaling to their own advantage through the expression of PD-L1 or PD-L2 on their surface, thus favoring the escape of immune surveillance [72].

PD-1 inhibitors can prevent the interaction of PD-1 with its ligands PD-L1 on tumor cells and inflammatory cells by binding to PD-1, leading to the restoration of antitumor activity of functionally depleted T cells. The first anti-PD-1 drug to be used for HCC was nivolumab, a fully human IgG4 monoclonal antibody that blocks PD-1 interactions with PD-L1 and PD-L2; now, several other PD-1 inhibitors are available for the treatment of HCC and are currently undergoing clinical trials, including pembrolizumab, tislelizumab, toripalimab and camrelizumab, which are all humanized IgG4 antibodies, and sintiliamab, a fully human IgG4 monoclonal antibody [73,74]. Different from PD-1 inhibitors, anti-PD-L1 drugs exert their anti-tumor efficacy by binding directly to the PD-L1 receptor on the surface of cancer cells rather than to PD-1 [75]. Durvalumab, a fully human IgG1 antibody, and atezolizumab, a humanized IgG1 antibody, are currently the most relevant anti-PD-L1 agents investigated in the field of HCC [73]; however, recently, avelumab, another IgG1 human antibody, has also demonstrated encouraging results in clinical trials with HCC patients [76]. Since the knowledge about the PD-L2 regulatory network is relatively ambiguous, there are no clinical trials about immunotherapy regimens against PD-L2 so far.

In theory, anti-PD-1 antibodies can block the binding of PD-1 to both its ligands (PD-L1 and PD-L2), whereas PD-L1 antibodies can only inhibit the binding of PD-1 to PD-L1 and, therefore, could be less effective. One meta-analysis of 19 randomized clinical trials involving more than 11.000 patients with cancer revealed a statistically significantly greater OS for patients treated with PD-1 inhibitors compared with patients treated with anti-PD-L1 drugs [77]. However, no patients with HCC were included in the analysis, thus further studies are required to confirm whether anti-PD-1 antibodies are associated with better outcomes compared to PD-L1 inhibitors also in HCC patients.

### 2.2. CTLA-4 Inibithors

CTLA-4 is an inhibitory co-receptor that is inducibly expressed on activated T cells and constitutively expressed on Tregs [78]. Due to its higher affinity, CTLA-4 competes with its homologous CD28 and binds to CD80/CD86 on the surface of APCs, transmitting an inhibitory signal that downregulates the function of T cells [79]. Additionally, CTLA-4 has been shown to lower levels of CD80/CD86 costimulatory molecules available on APCs through CTLA-4-dependent sequestration via transendocytosis [80]. Therefore, when the T cells are activated and the number of Tregs increases, as it occurs in the HCC microenvironment, the expression of CTLA-4 is up-regulated and the degree of T cell inflammatory response is reduced [81,82]. Contrary to PD-1/PD-L1 activity, however, the downregulation of T cells’ immune response occurs mainly in lymphatic tissues [75].

Any drugs that block CTLA-4 activity can counteract its immunosuppressive mechanism in the process of T cell activation, thus up-regulating the immune system and increasing its ability to recognize and destroy neoplastic cells [75]. The first CTLA-4 inhibitor investigated in the field of HCC was tremelimumab, an IgG2 human antibody; now, ipilimumab, an IgG1 human antibody, is also available for the treatment of HCC [83].

## 3. The Immune Modulation Effect of Locoregional Therapies

In several animal and human studies, locoregional treatments have been shown to induce immune responses in HCC patients, positively altering their tumor microenvironment [84,85]. The release of tumor antigens due to cell death and subsequent recruitment and activation of APCs and effector immune cells are the main processes responsible for the changes in anti-tumor immune responses after locoregional treatments [84].

Immunogenic cell death involves the translocation of calreticulin on the cell surface, the secretion of ATP, and the release of the non-histone chromatin protein high-mobility group box 1 (HMGB1) and other immunostimulatory molecules that collectively facilitate the recruitment and activation of APCs into the tumor microenvironment, the engulfment of tumor antigens from dying tumor cells and, finally, the optimal antigen presentation to T cells [85,86,87,88,89,90]. Locoregional treatments can induce both apoptosis and necrosis of tumor cells. Necrosis is a form of cell death characterized by loss of plasma membrane integrity, culminating in the escape of cell contents into the extracellular space, including tumor specific antigens, thus is known to be immunogenic; conversely, apoptosis is a programmed cell death in which the plasma membrane is not disrupted and cellular contents are packaged and then released into apoptotic bodies, thus it is regarded as immunologically “silent” [91,92]; nevertheless, previous reports have also implicated that certain types of apoptosis could be immunogenic and therefore favor the immune response against the tumor [93,94].

A plethora of cytokines, chemokines, and inflammatory/cell stress molecular markers have been described following the execution of the majority of locoregional treatments for HCC, supporting the immune modulation effect of these techniques. The effect of MWA as a single therapy was one of the first to be investigated, demonstrating the activation of Tregs, CD4+ and CD8+ T cell and NK cells, as well as the release of IL-12 [95,96]. The evidence that ablative therapy can cause tumor-specific immune responses was observed also in patients who underwent RFA, which can increase the number of tumor-associated antigen-derived peptides in peripheral blood [97], induce APCs activation and proliferation [98] and stimulate the secretion of Th1 cytokines (such as IL-2, TNF-α and IFN-γ) that promote CD8+ T activity [99]. Similarly, also TACE was reported to promote immunogenic cell death, as testified to by the increased serum levels of immunogenic cell death biomarkers following the procedure [100]; moreover, TACE can also promote Th17 and CD8+ activation and reduce the number of Tregs [101,102]. More recently, infiltration of CD8+ T cells and NK cells and an increase in cytokines levels (especially IL-1, IL-6 and IL-8) was found after TARE with yttrium-90 [103,104,105].

Locoregional therapy can promote systemic immune response by releasing neoantigens into blood circulation, but their effect alone might be too modest to prevent tumor recurrence and metastasis, even after successful treatments. Moreover, especially when incomplete, locoregional treatments can also induce immunosuppressive factors (such as IL-6, VEGF, HIF-1α, TGF-β, PD-1 and PD-L1), stimulate the accumulation of Tregs in the tumor and cause lymphopenia, leading to tumor progression in the end [106,107,108,109,110]. Incomplete T cell restoration despite antigen clearance and immune-tolerant liver environment might also affect the attenuation of immune surveillance. Additionally, their immunological effects appear limited in time. Indeed, as demonstrated by a previous study, the memory phenotype and lifetime of tumor-specific T cells were not sufficient to prevent HCC recurrence completely after RFA [97]. For all these reasons, the efficacy of locoregional treatments could be enhanced by their combination with immunotherapeutic drugs, which would guarantee the achievement of an immunologically more favorable tumor microenvironment [111,112]; at the same time, through a mutually beneficial and synergistic mechanism, the positive alteration of the tumor microenvironment derived from locoregional treatments may enhance ICI therapy efficacy (Figure 3) [38].

To date, there is no direct comparison between the different ablation or intra-arterial techniques, therefore it is not known whether one technique is superior to the others in inducing tumor-specific immune response [113]. In a previous study, it was demonstrated that serum levels of Glypican-3, a carcinoembryonic antigen inducing tumor-specific activation of cytotoxic T cells, were increased in 55% of patients with HCC after RFA and in 44% of patients after TACE, although these results were non-significant [114]. Interestingly, more recent evidence seems to suggest that TACE may have a greater immunogenic role than other locoregional treatments, possibly due to the potential immunogenic cell death induced by doxorubicin [115]. Doxorubicin is the most used chemotherapeutic agent for TACE and, despite the absence of a proven superiority compared to other drugs (such as cisplatin, epirubicin and mitomycin), is the only one to have demonstrated to possess immunogenic properties and thus trigger a significant tumor-specific immunological response [116]. In particular, anthracyclines such as doxorubicin seem to cause the post-transcriptional translocation of calreticulin from the endoplasmic reticulum, where it is involved in the maintenance of Ca^2+^ homeostasis, to the plasma membrane of tumor cells; surface-exposed calreticulin then acts as an “eat me” signal for phagocytosis by neighboring APCs, which is required for subsequent antigen cross-presentation to cytotoxic T cells [117]. Because chemotherapy is an integral part of TACE, these studies indicate that not only the immunogenic effects of embolization must be considered, but also the immune effects of the chemotherapy of choice. Therefore, if TACE is combined with immunotherapy, doxorubicin likely would lead to better outcomes compared to other chemotherapeutic agents.

## 4. The Current Evidence from Clinical Trials

The high risk of local and distant recurrence after locoregional treatments indicates the need for efficient adjuvant strategies to improve cure rates, even at very early and early stages. Features, such as large tumors, multinodularity, and vascular invasion (macroscopic or microscopic), are significantly related to higher recurrence rates in both ablative and intra-arterial therapies [118,119,120]. With this perspective, the addition of immunotherapy after locoregional treatments could amplify the effect of these treatments against micro-metastatic residual disease, especially in patients with a high risk of recurrence or those who would present clinical or hepatic deterioration after treatment. Similarly, there is a rationale to integrate immunotherapy in the neoadjuvant setting as well, especially in intermediate and advanced stages. The pre-treatment administration of ICIs can indeed leverage the higher levels of tumor antigens and thus promote the expansion of tumor-specific T cells, increasing the chance of cure following locoregional treatments [121,122].

One of the first trials that investigated the role of ICIs in combination with locoregional treatments in HCC patients evaluated the safety and efficacy of tremelimumab plus subtotal conventional TACE, RFA or cryoablation in patients who were non-responders to sorafenib. In particular, the protocol was shown to be safe and feasible, with no clear trends in adverse events or dose-limited toxicity; moreover, this therapeutic combination resulted in objective tumor responses even outside of the ablated or embolized zone, indicating that the systemic effects brought by locoregional therapies indeed exist [123]. The combination of tremelimumab plus ablation (RFA or cryoablation) or drug-eluting beads TACE (DEB-TACE) was also assessed in another study with HCC patients progressed on sorafenib therapy, proving the safety and efficacy of the protocol; in particular, the primary lesion kept shrinking and almost disappeared at 6 months and the untreated other intrahepatic lesions reduced in size gradually [84]. The enhanced efficacy of anti-PD-1 and ablative combined therapy was later confirmed in another retrospective study, where patients who underwent RFA plus camrelizumab or sintilimab demonstrated a longer OS and a higher recurrence-free survival (RFS) compared to those treated with RFA alone (32.5% vs. 10.0% and 51 weeks vs. 47.6 weeks, respectively) [124]. Similarly, a proof-of-concept clinical trial enrolling 50 patients with advanced HCC after sorafenib failure reported that additional RFA or MWA to anti-PD-1 therapy (nivolumab or pembrolizumab) increased the response rate from 10% to 24%. This latter study, moreover, documented that repeated ablations were also proved feasible and safe, reporting only common ablation-related complications that were easily managed as per the standard of care [125].

Three different studies [126,127,128] indicated that anti-PD-1 therapy (camrelizumab) plus TACE regimen is effective and safe, with effective tumor control, improved survival and manageable ICI-related adverse effects, leading to better outcomes than treatment with anti-PD-1 inhibitors alone; moreover, a longer interval between camrelizumab administration and TACE was related to the unsatisfying OS, whereas the timing of administration (before or after TACE) did not significantly influence the results. However, another study reported similar efficacy of TACE combined with camrelizumab compared to TACE alone, although the protocol was safe and tolerable [129]. Among the most common adverse events, itching was the most common, and is often associated with dermatitis and increased liver transaminases; whereas the appearance of colitis, thyroiditis and pneumonia is rarer. An interesting study compared the efficacy and safety of conventional TACE + camrelizumab with DEB-TACE + camrelizumab with the aim of determining which technique was superior. Despite both protocols being safe and well-tolerated, DEB-TACE produced better tumor response and PFS (70.4% vs. 40.7% and 10 vs. 3 months, respectively); however, these results could have been influenced by the inclusion of patients with large and multiple HCCs, who are theoretically more susceptible to this type of intra-arterial procedure; thus, further studies are needed [130].

Similar to TACE, even TARE in combination with nivolumab was demonstrated as a safe and effective treatment for HCC patients, showing a higher objective response rate (ORR) compared to both TARE alone and anti-PD-1 agents alone (30.6% vs. 20% vs. 15–23%, respectively) [131]; of note, the ORR in patients without extrahepatic spread was 43.5%, suggesting that TARE followed by nivolumab should be further evaluated in patients with BCLC B or BCLC C with no extrahepatic spread. One small retrospective trial examined patients with advanced HCC but preserved liver function who had received TARE and nivolumab with or without ipilimumab, documenting the safety of this association; moreover, there were no differences in toxicities between patients who received both therapies within 30 days of each other and those who received both therapies within 30–90 days [132]. The safety and efficacy of TARE plus anti-PD-1 therapy were also confirmed in other studies [133,134].

Despite this encouraging evidence, larger and comparative studies are needed to confirm the efficacy of immunotherapy combined with locoregional treatments in HCC patients. Currently, several other trials are exploring the role of ICIs in combination with locoregional treatments in HCC patients, with or without other drugs (such as tyrosine-kinase inhibitors), but participants are still being recruited or are receiving intervention, or data have yet to be analyzed. The role of numerous immunotherapeutic drugs is being tested in the adjuvant setting of patients who underwent ablative therapies, including nivolumab (the CheckMate 9DX trial, NCT03383458), atezolizumab plus bevacizumab (the IMbrave050 trial, NCT04102098) and pembrolizumab (the KEYNOTE-937 trial, NCT03867084); similarly, the use of nivolumab in both adjuvant and neoadjuvant settings after electroporation is currently being investigated (the NIVOLEP trial, NCT03630640). The number of studies that are evaluating the combination of immunotherapy with TACE and TARE in intermediate and advanced patients is even larger. The results of the LEAP-012 trial exploring TACE plus pembrolizumab plus lenvatinib in advanced HCC patients are eagerly awaited (NCT04246177), as are those of ongoing trials evaluating TACE plus atezolizumab plus bevacizumab (NCT04712643), TACE plus durvalumab plus bevacizumab (the EMERALD-1 trial, NCT03778957), TACE plus durvalumab plus bevacizumab plus tremelimumab (the EMERALD-3 trial, NCT05301842), TACE plus nivolumab (the TACE-3 trial, NCT04268888) and TACE plus nivolumab plus ipilimumab (the CheckMate 74W trial, NCT04340193). Similarly, the results that will emerge from trials combining TARE with yttrium-90 plus nivolumab (NCT03033446), pembrolizumab (NCT03099564) or durvalumab plus tremelimumab (the MEDI4736 trial, NCT04522544) are highly anticipated.

## 5. Current Challenges and Limitations of Combined Immunotherapy and Locoregional Therapies

The greatest challenge in investigating this combination approach still lies in the design of the clinical trials, specifically in the selection of an appropriate target population, proper control arms and adequate primary endpoints [135]. Moreover, the heterogeneity of both population and tumor burden should be considered before randomization since it can potentially limit the results [136]. For example, the different outcomes between virus-related and non-virus-related HCC observed in other trials of immunotherapies seem to suggest that this element should be incorporated as a stratification factor in addition to the geographical region [137]. Furthermore, the optimal regimens of locoregional treatment (dose/fraction of radiation therapy, types of chemotherapeutic agents, etc.) that will best induce immunogenic cell death and the timing and sequence of both locoregional treatments and immunotherapy should also be addressed [138]. Finally, there are still methodologic differences in how these trials assess treatment response since they combine agents that require iRECIST with therapeutical procedures requiring mRECIST, adding a complexity that remains to be determined.

Besides these “theoretical” challenges, however, there are also several “practical” issues in implementing this combination therapy in clinical practice. For example, the Italian Liver Cancer (ITA.LI.CA) group has shown that in real-life practice, due to the numerous restricted selection parameters, only 10–20% of HCC patients are eligible for first-line ICI therapy and this percentage is reduced to <10% in the second-line treatment. Therefore, considering that about 30–40% of them do not respond to these agents, only a small number of HCC patients could actually benefit from immunotherapy [139]. Moreover, the contraindications and the feasibility of locoregional treatments in these patients should also be acknowledged, since they are not negligible [140].

Therefore, there is an urgent need for effective predictive serological and/or tissue biomarkers to identify patients likely to benefit from immunotherapy and thus dictate patient-specific therapy choices and reduce the economic burdens on healthcare systems; in addition, it would be possible to avoid ICI-associated adverse events in those patients identified as non-responders. PD-L1 expression is widely used today for the selection of anti-PD-1 therapy in patients with non-small cell lung cancer and melanoma [141,142]; as for HCC, this association has not yet been sufficiently investigated and PD-L1 expression cannot be considered a binary marker to help decide which patients should receive anti- PD-1 therapy. In addition, a study revealed a significant correlation between tumor mutation burden (TMB) and clinical outcomes after PD-1 inhibitors, implying that tumors with high TMB would present a greater number of tumoral neoantigens and thereby would have a greater chance of being recognized by tumor-specific T cells [143]; however, since HCC proved to be less immunogenic compared to other tumors and showed a low TMB (approximately 5 Mut/Mb), the role of this biomarker in these patients remains limited [144]. More recently, overexpression of TIM-3 and LAG-3 as well as Wnt/β-catenin mutations have emerged as possible biomarkers of response in patients receiving anti-PD-1 therapies; nonetheless, data are still preliminary and further confirmations are necessary before drawing firm conclusions [145,146,147]. Therefore, to date, no validated molecular and/or genetic biomarkers predicting response to ICIs in patients with HCC have been identified.

Finally, there is a unique and challenging subset of patients that deserves special attention, i.e., those in whom HCC recurs after liver transplantation. In fact, despite its proven curative efficacy, the recurrence rate of HCC following transplantation is still 15–20%. [148] However, organ transplantation has typically been an exclusion criterion in every clinical trial testing ICIs since immunotherapy, through the activation of T cells, can cause allograft rejection and, subsequently, lead to end-stage organ failure in a high percentage of subjects (37.5% and 75%, respectively) [149]. Moreover, due to the immunosuppressive status, the efficacy of immunotherapy could be reduced because ICIs require competent T-cell populations to exert their antitumor effects [150]. Therefore, until further experience is provided, the use of ICI should be avoided in the neoadjuvant setting for patients on the liver transplant waiting list as well as in the post-transplant setting, and sorafenib or lenvatinib should remain the first-line treatment for this subgroup of HCC patients.

## 6. Future Perspectives and Promises of Combined Immunotherapy and Locoregional Therapies

Despite that the therapeutic combination of ICIs and locoregional treatments appears promising, its antitumor efficacy may be impaired by the hypoxic mechanisms secondary to locoregional approaches, which might increase the level of pro-angiogenetic cytokines (such as VEGF-1, VEGF-2, TGFβ) and, therefore, promote tumor angiogenesis and metastasis development [38]. Moreover, hypoxia can reactivate APCs, promote the activation, infiltration and migration of lymphocytes and reduce the recruitment of inhibitory cells such as Tregs and MDSCs [151]; furthermore, PD-L1 expression is strongly dependent on transcriptional regulation of hypoxia-inducible factor 1 alpha (HIF1α) [152]. Based on these premises and the approval of the combination of atezolizumab and bevacizumab as first-line treatment for unresectable HCC [41], several studies are currently investigating the synergistic effects of locoregional therapies with immunotherapy and antiangiogenic agents with the hope to extend the clinical benefit of this combination. Therefore, VEGF/VEGFR inhibitors could have a double effect on cancer cells, counteracting the paradoxical angiogenic effect of locoregional therapies and boosting immunity through parallel and distinct mechanisms, thus further priming tumors for immunotherapy and leading to a stronger immune response. Some studies have shown the effectiveness and safety of TACE combined with antiangiogenic therapy and immunotherapy in advanced HCC [153,154,155]. In addition, TACE combined with antiangiogenic therapy and immunotherapy has been demonstrated to remarkably improve OS and PFS over both the combined antiangiogenic and immunotherapy or the combined TACE and antiangiogenic therapy in unresectable HCC patients [156,157,158,159]. Similar results have also been observed with other locoregional approaches, including DEB-TACE [160] and MWA [161].

Although the preliminary outcomes of these therapeutic combinations are encouraging, countless other treatment possibilities can still be explored. For example, radioimmunotherapy has recently emerged as a valid therapeutic option for HCC. In particular, this technique offers a selective internal radiation therapy approach using radionuclides conjugated with tumor-specific monoclonal antibodies, thus combining immunotherapy and radiotherapy. In particular, by binding to the cancer cell surface, these radioimmunoconjugates enable a targeted concentration of radiation in tumor tissue, leading to DNA damage and finally causing tumor cell death. Following this principle, iodine-131 labeled metuximab (also commercially known as Licartin) gained approval for clinical therapy of primary HCC from the China State Food and Drug Administration [162]. This radioimmunoconjugate directly targets HAb18G/CD147, a cell antigen with multiple functions highly expressed on HCC cells. In addition, to allow for the concentration of radionuclides in HCC tissues, iodine-131 metuximab can directly impair the adhesion and motion of tumor cells; furthermore, it can block metalloproteinases production and VEGFR-2 phosphorylation, thus effectively inhibiting HCC growth and metastasis [163,164]. A combination of iodine-131 metuximab with TACE has recently demonstrated an improved clinical efficacy in HCC therapy compared to TACE alone, in terms of both tumor response and OS, with a similar tolerability profile [165,166]. Theoretically, the arterial embolization activity of TACE can indeed enhance the anti-tumor effects of iodine-131 metuximab by reducing tumoral blood flow and resulting in tumor retention of the radioimmunoconjugate; moreover, retention of the anti-cancer drug in the tumor may have a radiosensitizing effect and iodine-131 metuximab can eliminate residual cancer cells after TACE for its continuous radiation. Interestingly, one randomized trial reported that also the combination of iodine-131 metuximab with RFA resulted in improved outcomes for patients at very early, early and intermediate stages, with greater anti-recurrence benefit than RFA alone; however, this significant improvement was not detected in the CD147-negative subpopulation, thus further analysis is still needed [167]. Therefore, radioimmunotherapy in combination with TACE or RFA and, hypothetically, even with systemic therapies (including sorafenib and other VEGF/VEGFR inhibitors) could certainly represent a promising field of investigation in the near future.

Adoptive cell therapies represent a novel approach to HCC therapy. In this approach, autologous immune cells are extracted from the patient, activated and expanded ex vivo, then reinfused into the patient; these can include cytokine-induced killer (CIK) cells, γδ T cells, dendritic cells, NK cells, lymphokine-activated killer (LAK) cells and tumour-infiltrating lymphocyte (TILs). Moreover, genetically modified immune cells, including chimeric antigen receptor T cells (CAR-T) directed against glypican-3 (GPC3), are currently in development [168,169]. Despite that it is too early for robust results in the HCC setting, it is worth mentioning that the few trials available in the literature seem to confirm that the combination of adoptive cell therapies with local ablation or TACE can be exploited to augment therapeutic efficacy and prevent tumor recurrence [170,171]. Recent data suggest that CIK cells transfusion therapy combined with TACE and/or RFA treatment is associated with longer OS and PFS compared to TACE and/or RFA treatment alone [172]; similarly, even the combination of locoregional treatments with NK cells, γδ T cells and CAR-T cells was reported to be safe and showed encouraging clinical efficacy [173,174,175]. With the identification of a growing number of HCC-associated antigens, tumor vaccines which increase specific immune responses to tumor antigens as well as oncolytic viruses—that can selectively replicate in tumor cells and damage them without harming normal cells—are being investigated and developed [176]. Unfortunately, clinical trials based on adoptive cellular therapies, tumor vaccines and oncolytic vaccines are relatively few compared to those studying ICIs, probably due to the limitations of in-house cell therapy facilities and the obvious higher costs [63,177].

Finally, multiple studies have revealed that qualitative and quantitative alterations of the gut microbiota play an important role in HCC pathogenesis. Chronic liver diseases, indeed, are associated with an imbalance in bacterial composition and metabolic activities (dysbiosis) as well as with changes in the intestinal barrier leakiness, which altogether lead to hepatic exposure to bacterial metabolites and microbiota-associated molecular patterns (MAMPs); additionally, in patients with chronically injured liver, hepatic detoxification, degradation, and clearance are compromised, thus hepatic exposure to this gut-derived microbial toxicity is further enhanced. Both bacterial metabolites and MAMPs promote hepatocarcinogenesis via multiple mechanisms, supporting a chronic inflammatory state, favoring fibrosis development and inducing senescence of hepatic stellate cells [178,179]. The gut microbiome seems to also have a notable impact on responses to immunotherapy in HCC patients. For example, an interesting study showed that fecal samples from patients responding to PD-1 inhibitors presented higher taxa richness compared to fecal samples of non-responders [180]; similarly, a significant microbial dissimilarity was observed in fecal bacteria between patients with unresectable HCC who responded to immunotherapy and patients who did not respond to therapy [181]. Therefore, considering the potential modulatory role of human microbiota on antitumoral immune response, the additional administration of specific probiotics, as well as the more complex fecal microbiota transplantation in non-responder patients are currently being investigated, with rather promising preliminary results. Preclinical studies have demonstrated that supplementation of certain commensal bacterial genera (such as *Akkermansia muciniphila*) can restore responsiveness to immunotherapy in mice with melanoma [182]. More recently, fecal microbiota transplantation from responder donors in patients with metastatic melanoma, refractory to anti-PD-L1 therapy, led to a clinical response and tumor regression in 3 out of 10 subjects; moreover, all 3 participants that responded to immunotherapy following transplantation received samples from the same donor, indicating the choice of donor stool may be critical in inducing sensitivity to immunotherapy [183]. To date, however, there are no available studies that have analyzed the associations between microbiota alterations and locoregional treatments. In the future, with the increasing possibility of acting on the gut bacterial composition, targeted studies investigating this theme should be considered.

## 7. Conclusions

Locoregional treatments have been shown to positively alter the tumor immune microenvironment of HCC; therefore, the association of locoregional treatments and immunotherapy could contribute to increasing their efficacy through a synergistic mechanism, thus improving the survival of HCC. Despite the encouraging results, however, many unanswered questions still remain, including which immunotherapy and locoregional treatment can guarantee the best survival and clinical outcomes, the most effective timing and sequence to obtain the most effective therapeutic response and which biological and/or genetic biomarkers can be used to identify patients likely to benefit from these combined approaches. Therefore, when planning new clinical studies, it is essential to homogenize, as far as possible, the trial designs to objectively evaluate the clinical contribution of this association. Finally, to maximize the results of the combined anti-tumor strategy, it is pivotal to enhance the multidisciplinary cooperation between hepatologists, radiologists, nuclear doctors and oncologists.

## Figures and Tables

**Figure 1 ijms-24-08598-f001:**
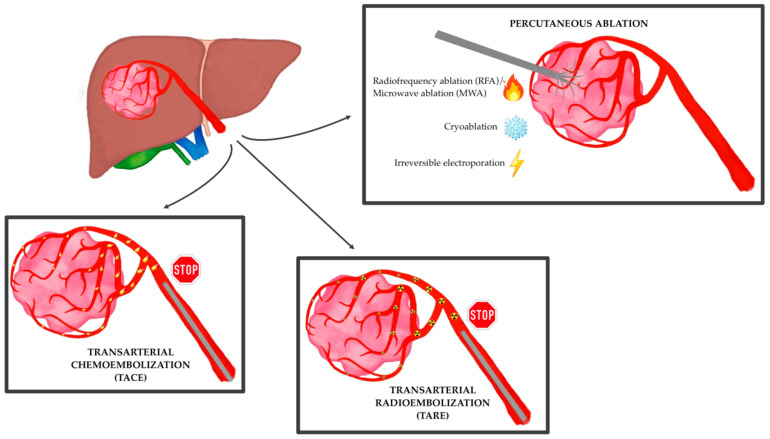
The main locoregional techniques for the treatment of hepatocellular carcinoma.

**Figure 2 ijms-24-08598-f002:**
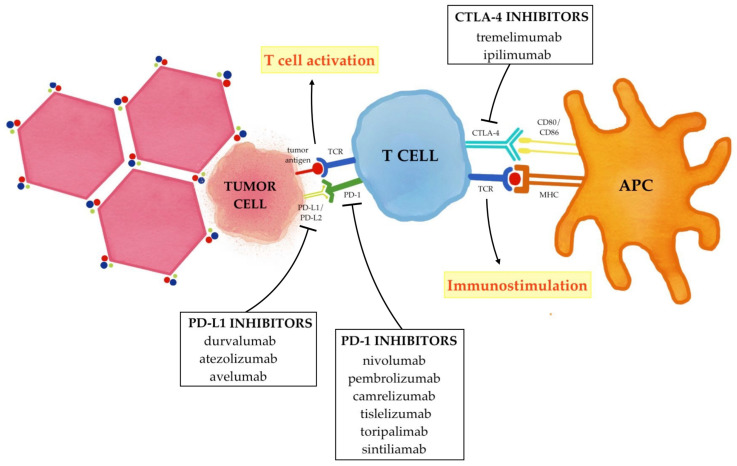
Immune checkpoint inhibitors in hepatocellular carcinoma. PD-1 binding its ligand PD-L1/PD-L2 prevents TCR signaling, blocks T cell proliferation, and induces the exhaustion of T cells. CTLA-4 binds CD80/CD86 and blocks the activation of the T cells. The inhibition of these immune checkpoints with PD-1/PD-L1 inhibitors and CTLA-4 inhibitors promote T cell activation and up-regulate the immune system, thus reactivating the anticancer immune response.

**Figure 3 ijms-24-08598-f003:**
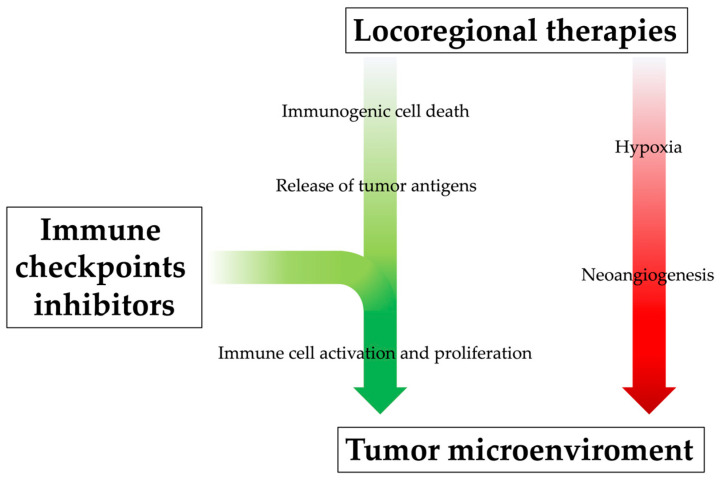
The rationale behind combining locoregional therapies and immunotherapy. Locoregional therapies, especially when incomplete, can increase the level of pro-angiogenetic cytokines and thus promote neoangiogenesis of residual cancer cells and metastasis development; at the same time, however, they promote systemic immune response by releasing neoantigens into blood circulation, although this immunogenic effect might be too modest. The immunological efficacy of locoregional treatments could be enhanced by their combination with immunotherapeutic drugs, which would promote immune cell activation and proliferation, positively influencing the tumor microenvironment.

## Data Availability

Not applicable.

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
