# Peer review of "The Synergistic Effect of Interventional Locoregional Treatments and Immunotherapy for the Treatment of Hepatocellular Carcinoma"

_ijms, 2023, doi:10.3390/ijms24108598_

Round 1

Reviewer 1 Report

In the present review article, Brandi et al describe current application of immunotherapy in combination with locoregional therapies for the treatment of HCC and provides a critical evaluation of the current status and future directions. This review article provides a detailed information to the clinician on the combination therapy for the treatment of the HCC patients. There are some issues which needs to be addressed

11. As it’s a review article, it would be better to compare the efficacy of a single agent and combination of locoregional therapy and immunotherapy in HCC in a tabular format.

22. What is the rationale behind combining the locoregional therapy and immunotherapy? It should be addressed diagrammatically in the paper.

33.References from recent publications are not cited in the paper:

- Zheng et al in current oncology reports entitled Emerging Opportunities for Combining Locoregional Therapy with Immune Checkpoint Inhibitors in Hepatocellular Carcinoma.

-  Dendy et al in Live cancer entitled Locoregional Therapy, Immunotherapy and the Combination in Hepatocellular Carcinoma: Future Directions

Thus, I strongly encourage the authors to discuss above mentioned review article in the paper.

44. It would be better to mention few clinical trials ongoing evaluating the efficacy of the combination of locoregional therapy and immunotherapy in HCC in the paper.

Quality of English language are fine

Author Response

Dear Reviewer,

Please find enclosed the revised version of our manuscript entitled “The synergistic effect of interventional locoregional treatments and immunotherapy for the treatment of hepatocellular carcinoma.

Thank you for the opportunity to revise and improve our paper according to your comments, with the aim to publish it in International Journal of Molecular Sciences.

We have modified the main text in accordance with your insightful and significant suggestions and we have replied point by point to all requested revisions.

We hope that now our manuscript reaches a suitable level for a possible publication.

Dear Authors,

In the present review article, Brandi et al describe current application of immunotherapy in combination with locoregional therapies for the treatment of HCC and provides a critical evaluation of the current status and future directions. This review article provides a detailed information to the clinician on the combination therapy for the treatment of HCC patients. There are some issues which needs to be addressed.

  1. As it’s a review article, it would be better to compare the efficacy of a single agent and combination of locoregional therapy and immunotherapy in HCC in a tabular format.

RE: Dear Reviewer, thank you for your appreciation and insightful comments. The present review aims to summarize the current application of immunotherapy in combination with locoregional therapies for the treatment of HCC from a general point of view and, therefore, it is not a meta-analysis aimed to compare the efficacy of one drug vs. another in the setting of combined immune and loco-regional therapy. Moreover, as we described in section “5. Current challenges and limitations of combined immunotherapy and locoregional therapies” the heterogeneity of both study population and tumor burden, as well as the significative differences in the study design and therapeutic regimens(dose/fraction of radiation therapy, types of chemotherapeutic agents, etc.) limits an effective comparison between different therapeutic approaches. Therefore, if the Editor agrees with us, we would prefer to try to address the broader topic of combined immunotherapy and loco-regional approaches, without focusing only on one drug.

  1. What is the rationale behind combining locoregional therapy and immunotherapy? It should be addressed diagrammatically in the paper.

RE: Dear Reviewer thank you very much for the very suitable suggestions. We have now included a new figure which diagrammatically addresses the rationale behind combining locoregional therapy and immunotherapy (Figure 3).

33.References from recent publications are not cited in the paper:

- Zheng et al in current oncology reports entitled Emerging Opportunities for Combining Locoregional Therapy with Immune Checkpoint Inhibitors in Hepatocellular Carcinoma.

-  Dendy et al in Live cancer entitled Locoregional Therapy, Immunotherapy and the Combination in Hepatocellular Carcinoma: Future Directions

Thus, I strongly encourage the authors to discuss above mentioned review article in the paper.

RE: Dear Reviewer, thank you very much for the very suitable suggestions. We have now included the recommended references.

  1. It would be better to mention few clinical trials ongoing evaluating the efficacy of the combination of locoregional therapy and immunotherapy in HCC in the paper.

RE: Dear Reviewer, thank you very much for the helpful suggestion. We agree with you that mentioning few ongoing clinical trials evaluating the efficacy of the combination of locoregional therapies and immunotherapy could improve our paper. As such, we have added a brief mention of the most promising ones at the end of the section “4. The current evidence from Clinical Trials”.

We look forward to hearing from you at your earliest convenience.

Sincerely,

Nicolò Brandi and Matteo Renzulli

Department of Radiology, IRCCS Azienda Ospedaliero-Universitaria di Bologna, Via Albertoni 15, Bologna, Italia.

Reviewer 2 Report

In their review, Bardi et al. investigate the synergistic effect of interventional locoregional treatments and immunotherapy for the treatment of hepatocellular carcinoma.  In general the manuscript is well written, however  certain parts require clarification. 

Suggestions:

1. Figure 2 should be check again.

2. The discussion part should be shortened and strenghtened.

3. Some references need to be updated.

Thank you very much.

Author Response

Dear Reviewer,

Please find enclosed the revised version of our manuscript entitled “The synergistic effect of interventional locoregional treatments and immunotherapy for the treatment of hepatocellular carcinoma.

Thank you for the opportunity to revise and improve our paper according to your comments, with the aim to publish it in International Journal of Molecular Sciences.

We have modified the main text in accordance with your insightful and significant suggestions and we have replied point by point to all requested revisions.

We hope that now our manuscript reaches a suitable level for a possible publication.

In their review, Brandi et al. investigate the synergistic effect of interventional locoregional treatments and immunotherapy for the treatment of hepatocellular carcinoma.  In general, the manuscript is well written, however certain parts require clarification.

RE: Dear Reviewer, thank you very much for the positive comments and helpful suggestions to improve our manuscript further

Suggestions:

  1. Figure 2 should be check again.

RE: Dear Reviewer, thank you very much for the helpful suggestion. We have carefully re-read the figure’s legends to fix the typing errors.

  1. The discussion part should be shortened and strenghtened.

RE: Dear Reviewer, thank you very much for the very nice comment about the discussion of our manuscript. We have carefully reread it and strengthened it as suggested.

  1. Some references need to be updated.

RE: Dear Reviewer, thank you very much for the very suitable suggestions. We have checked all the references and, as suggested also by Reviewer 1, have included some new ones which have been recently published.

We look forward to hearing from you at your earliest convenience.

Sincerely,

Nicolò Brandi and Matteo Renzulli

Department of Radiology, IRCCS Azienda Ospedaliero-Universitaria di Bologna, Via Albertoni 15, Bologna, Italia.